# Rietveld Quantitative Phase Analysis of Oil Well Cement: In Situ Hydration Study at 150 Bars and 150 °C

**DOI:** 10.3390/ma12121897

**Published:** 2019-06-12

**Authors:** Edmundo Fraga, Ana Cuesta, Jesus D. Zea-Garcia, Angeles G. De la Torre, Armando Yáñez-Casal, Miguel A. G. Aranda

**Affiliations:** 1ALBA Synchrotron, Carrer de la Lum, 2-26, Cerdanyola del Vallès, 08290 Barcelona, Spain; efraga@cells.es; 2Departamento de Ingeniería Industrial, Universidade da Coruña, Ferrol, 15403 A Coruña, Spain; armando.yanez@udc.es; 3Departamento de Química Inorgánica, Cristalografía y Mineralogía, Universidad de Málaga, 29071 Málaga, Spain; a_cuesta@uma.es (A.C.); jdavidzea@uma.es (J.D.Z.-G.); mgd@uma.es (A.G.D.l.T.)

**Keywords:** high-pressure equipment, powder diffraction, synchrotron radiation, cement hydration, reactivity, oil well cement

## Abstract

Oil and gas well cements are multimineral materials that hydrate under high pressure and temperature. Their overall reactivity at early ages is studied by a number of techniques including through the use of the consistometer. However, for a proper understanding of the performance of these cements in the field, the reactivity of every component, in real-world conditions, must be analysed. To date, in situ high energy synchrotron powder diffraction studies of hydrating oil well cement pastes have been carried out, but the quality of the data was not appropriated for Rietveld quantitative phase analyses. Therefore, the phase reactivities were followed by the inspection of the evolution of non-overlapped diffraction peaks. Very recently, we have developed a new cell specially designed to rotate under high pressure and temperature. Here, this spinning capillary cell is used for in situ studies of the hydration of a commercial oil well cement paste at 150 bars and 150 °C. The powder diffraction data were analysed by the Rietveld method to quantitatively determine the reactivities of each component phase. The reaction degree of alite was 90% after 7 h, and that of belite was 42% at 14 h. These analyses are accurate, as the in situ measured crystalline portlandite content at the end of the experiment, 12.9 wt%, compares relatively well with the value determined ex situ by thermal analysis, i.e., 14.0 wt%. The crystalline calcium silicates forming at 150 bars and 150 °C are also discussed.

## 1. Introduction

Portland cement (PC) is the most manufactured product in the world, as it is the main component of the construction industry [1]. In this paper, cement nomenclature will be used for describing the cement phases: C = CaO, S = SiO_2_, A = Al_2_O_3_, F = Fe_2_O_3_, $\bar{S}$ = SO_3_, and H = H_2_O. PC is a multimineral material [2] containing ~65 wt% of C_3_S or alite; ~15 wt% of C_2_S or belite; ~15 wt% of C_4_AF or tetracalcium aluminoferrite, and ~5 wt% of C_3_A or tricalcium aluminate (ideal stoichiometries: Ca_3_SiO_5_, Ca_2_SiO_4_, Ca_4_Al_2_Fe_2_O_10_ and Ca_3_Al_2_O_6_, respectively). Additionally, calcium sulphates are added to regulate the setting of the pastes/mortars/concretes.

The hydration of PC basically consists of two set of reactions (that interact with each other): the silicate reactions and the aluminate reactions [3]. C_3_S is the most important phase in PC. Under normal conditions of pressure and temperature, the hydration reaction of C_3_S [4] consists of its dissolution and the precipitation of a nanocrystalline calcium–silicate–hydrate (C–S–H) gel, and crystalline portlandite, CH, according to reaction 1. C–S–H gel is a complex hierarchically arranged material with overall chemical stoichiometry close to (CaO)_1.8_SiO_2_·4H_2_O, which can be broken down at the nanoscale as [Ca_1.2_SiO_3.1_(OH)_2_·H_2_O]·[Ca(OH)_2_]_0.6_·[H_2_O]_2.3_ to describe its three main intermixed components [4]: [defective clinotobermorite]·[nano-sized calcium hydroxide layers]·[gel pore water. The reaction of belite is slower, yielding C–S–H gel and 0.2 moles of portlandite.
(1)$$C_{3}S+5.2H\to 1.2\mathrm{CH}+{\mathrm{Ca}}_{1.8}{\mathrm{SH}}_{4}$$

In addition, at ambient conditions, the aluminate reaction consists of the dissolution of aluminates, i.e., C_3_A and C_4_AF and calcium sulphate sources, to yield mainly crystalline ettringite (also known as AFt), according to reaction (2). In the absence of sulphates, monosulphoaluminate phases form like Kuzelite, Ca_4_Al_2_(OH)_12_SO_4_·6H_2_O.
(2)$$C_{3}A+3{C\bar{S}H}_{2}+26H\to C_{6}{A\bar{S}H}_{32}$$

On the other hand, Oil-Well-Cement (OWC) is a special kind of cement [5] used for specific applications mainly in oil, gas and geothermal industries. The pastes derived from OWC are subjected to high pressures (up to 1000 bar) and high temperatures (up to 300 °C) during curing. Consequently, the hydration processes changes substantially, and the hydration phases formed under these conditions are different (or they can be) than those obtained at room temperature and atmospheric pressure [6,7]. Based on previous studies [7,8,9], it is evident that the effect of temperature is larger than that of pressure in cement hydration. Although there are many studies of cement hydration at high temperature, there are not many in situ diffraction works focused on the combined role of pressure and temperature on cement hydration. Furthermore, these studies were qualitative or semi-quantitative as they mainly followed selected diffraction peaks [see for instance [10] and references therein]. These studies were mainly dedicated to the hydration of C_3_S [6,7,11,12].

High pressures increase the hydration rate [11] of a cement, mainly the hydration kinetics of C_3_S [6]. Moreover, it was found that the length of the induction period, very much related to the time required for initial setting, is reduced as the pressure and temperature increases [6,7,11]. However, high temperature and high pressure also lead to the formation of different hydrated products. For instance, it has been reported that C–S–H gel is not stable at high temperature [11] and it (partially) reacts/decomposes resulting in other (crystalline) hydrates, such as α-C_2_SH and/or Jaffeite, C_6_S_2_H_3_. An ex situ hydrothermal study [13] also showed that at high temperature the C–S–H and ettringite phases were decomposed into Jaffeite via α-C_2_SH and monosulphate, respectively. Furthermore, the formation of these hydrated phases under high pressure and temperature conditions leads to poor mechanical properties and pore structure degradation. Consequently, some authors [14,15] employed different additives, for instance, silica fume or blast furnace slag, to try to inhibit the crystallization of those non-desired crystalline calcium silicates hydrates. It seems that the use of additives lead to an improvement in the microstructure of these pastes resulting in the enhancement of mechanical properties [16]. However, much research is still required to clarify these observations and to establish proper correlations between phase development, microstructure and mechanical properties [17].

Synchrotron X-ray powder diffraction can be used to follow in real time the in situ hydration reaction [18] because it is possible to select an X-ray beam of high energy (which enables penetration of the cell component(s)) and with very high flux (which yields a good signal-to-noise ratio in the diffraction data). For this study, we have used a home-made high pressure and high temperature spinning capillary cell [19] which has been successfully operated up to 200 bars and 200 °C. This cell is an evolution of a previous one [20] but with a new design that allows the capillary to spin in order to increase particles statistics with the final goal to carry out quantitative phase analyses. This capillary cell makes it possible to collect in situ X-ray powder diffraction data of hydrating cements to be analysed by Rietveld methodology. The quality of a single Rietveld quantitative phase analysis was already checked in a previous study reporting the cell design and operation [19].

The main objective of this work was to demonstrate that in situ X-ray powder diffraction data collected from the new capillary cell is of sufficient quality to carry out quantitative phase analyses, and not only to follow selected diffraction peaks. This is illustrated by the study of the hydration of an OWC under 150 bars and 150 °C, where the in situ diffraction data have been analysed by the Rietveld method in order to understand the hydration kinetics and the phase development. The amounts of the initial phases and the formed components are reported up to 14 h of hydration.

## 2. Materials and Methods

### 2.1. Sample Preparation

A commercial OWC Class G, high sulfate resistant HSR, (Dyckerhoff-Lengerich, Lengerich, Germany) was used for this work. The elemental composition is given in Appendix A, as Appendix A. This cement has a Blaine fineness value of 340 m^2^/kg, the Brunauer–Emmett–Teller BET specific surface measured by N_2_ sorption was 0.88 (1) m^2^/g and the Particle Size Distribution (PSD) was measured using a laser analyser (Mastersizer S, Malvern, UK). The hydration of the OWC was performed by adding water to the powder sample at a water/cement mass ratio of 0.47. Then, the mixture was stirred by hand in a plastic beaker for 2 min. The paste was immediately loaded, with the aid of a syringe and a short piece of silicone tubing, into the sapphire capillaries (Saint-Gobain crystals) with outer and inner diameters of 3.18 mm and 1.75 mm, respectively. Polytetrafluoroethylene PTFE cylindrical plugs of 2 mm of length and 1.75 mm outside diameter (with tolerance for its width smaller than 0.1 mm) were used to block both ends of the sapphire capillaries at least at 10 mm from the end of the capillary. These flexible plugs allow the pressure from the oil system to be transmitted from both sides.

### 2.2. Laboratory Initial Characterization

Thermal analysis measurement of the final paste (after the synchrotron experiment) was performed in a SDT-Q600 analyzer (TA instruments, New Castle, DE, USA) under a flow of dry air. The initial and final setting times were determined from Vicat—methodology following UNE-EN 196-3:2005. The isothermal calorimetric study was performed in an eight channel Thermal Activity Monitor (TAM) instrument using glass ampoules. The heat flow was collected up to 3 days at 20 °C.

### 2.3. Synchrotron X-ray Powder Diffraction (SXRPD) Experiment

Full details about the beamline, capillary cell and experimental conditions have been very recently reported [19]. Here, we provide a summary for convenience. A photon energy of 20 keV (λ = 0.62278 Å) was selected with a Si (111) channel-cut monochromator to collect powder diffraction data in Debye-Scherrer configuration. The beam size was 0.8 mm vertical and 1.2 mm horizontal. The detector was a LX255-HS Rayonix CCD placed at 313 mm from sample (tilt vertical angle of 28.92°).

A sapphire capillary fill of quartz was used as the standard to calibrate the detector setup. The sealed sapphire capillary containing the paste was loaded into the spinning capillary cell. Firstly, the static pressure was manually generated by a pump generator to achieve 150 bars. Subsequently, the heating system was turned on to reach 150 °C. It has to be borne in mind that the pressure application took place 46 min after paste mixing and the desirable temperature was reached 20 min later. The initial 46 min of hydration occurred at room temperature and pressure. Consequently, it is considered here as initial time, t_0_, the time in which the desired pressure was reached, it means 46 min after initial water-cement mixing time. This criterion is followed in the text, tables and figures.

The cell was rotated at 240 rpm. It contains a Micos LS-180 translation stage that enables to collect 2D data in any desired horizontal point of the capillary. To ensure that representative datasets are taken, snapshots at 5 different positions along the capillary, at 0.5 mm intervals, were acquired with an exposure time of two seconds per snapshots. The first hydrating in situ pattern was collected 9 min after reaching the selected pressure (i.e. during heating). Then, patterns were collected with intervals of ~15 min for 14 h. The 2D images were reduced to 1D data by pyFAI software (version: 0.10.3, European Synchrotron Radiation Facility, ESRF, Grenoble, France) [21] and the five 1D raw patterns, collected at different capillary positions, were sum up with a local software yielding the final dataset to be analysed by Rietveld methodology.

### 2.4. SXRPD Data Analysis

The regions of the powder diffraction patterns which include the Sapphire diffraction peaks were excluded for the fits. Generalized Structure Analysis System (GSAS) suite of programs [22] were used to analyse all powder patterns to obtain Rietveld quantitative phase analyses (RQPA). The references for the crystal structures used to calculate the powder patterns are given in Table 1. Final global refined parameters were background coefficients, zero-shift error, cell parameters and peak shape parameters using a pseudo-Voigt function [23].

## 3. Results and Discussion

### 3.1. Initial OWC Characterization

As expected for a G-type OWC, the alkaline content (K_2_O and Na_2_O) was smaller than 0.8 wt%, see Appendix A. The RQPA for this cement was previously reported [19]. Also, as expected, the C_3_A content measured by Rietveld method from X-ray laboratory data was low. On the other hand, fineness is as important as the phase composition to understand the kinetic of the reactions at early ages as these strongly depends upon the particle sizes. Therefore, the PSD curve for this OWC is shown in Appendix A. The average particle size, dv, 50, was 13 µm and dv, 90, was 38 µm.

In order to gain further insight into the early age reactivity of this cement, some ambient tests were carried out. It was not possible to determine the setting time for a paste with a water-to-cement (w/c) mass ratio of 0.47 due to bleeding. Therefore, a second paste with w/c = 0.35 was used for the Vicat measurement. For the paste with w/c mass ratio of 0.35, the initial setting was 388 min and the final setting was slightly larger than 550 min. A calorimetric study was also carried out for these two pastes (w/c ratios of 0.35 and 0.47). Appendix A displays the calorimetric data showing the maxima of the heat flow at 13 and 18 h for the pastes with w/c ratios of 0.35 and 0.47, respectively. These data were collected at 20 °C and room pressure and much faster kinetics are expected at 150 bars and 150 °C, see below.

### 3.2. Qualitative Study

Appendix A displays 2D raw patterns for the initial (anhydrous) OWC, the paste hydrated at 1 h and 38 min and at 11 h and 22 min, as representative examples. Five of these snapshots were summed to give a representative 1D SXRPD pattern at a given hydration time. Figure 1 displays a 3D view of the one-dimensional SXRPD patterns collected every 15 min at 150 bars and 150 °C. The qualitative phase evolution can be derived from the evolution of the diffraction peaks. This is useful for quantitative analyses of estimations of the appearance of new phases, as described in the next section. From this plot can be stated that alite has almost fully reacted within the first 7 h of hydration.

A better look for the evolution of the component phases can be obtained by selecting appropriated time windows. Figure 2 shows 1D SXRPD patterns for OWC pastes during the first 2 h of hydration at 150 bars and 150 °C. In the pattern collected at 26 min, the diffraction peaks from gypsum vanished and the main diffraction peak of bassanite (5.9° [2θ/λ = 0.62278 Å]) appeared, indicating the partial dehydration of gypsum in these conditions. At this early reaction time it can be also observed that the AFt reflection, located at 3.7°, starts to decrease and vanished about 38 min. This is mainly due to the temperature as ettringite decomposes at temperatures above ~80 °C. Moreover, katoite starts to precipitate at very early hydration ages, after 26 min in these conditions. Figure 2 also shows that alite starts to dissolve close to 26 min but the diffraction peaks of crystalline portlandite are barely visible at this hydration age. The portlandite appearance can be inferred from its diffraction peak located at 7.2°, which is clearly visible at 32 min of hydration; its formation accelerates after one hour.

Figure 3 shows 1D SXRPD patterns for OWC pastes for up to 14 h of hydration. In this figure, it can be highlighted the formation of crystalline hydrated phases that are not common at room temperature and pressure, such as α-C_2_SH and Jaffeite. It seems that the formation of α-C_2_SH starts close to 7 h of hydration and the formation of a very small amount of Jaffeite takes place after 11 h of hydration. It is also worth noting that C_2_S phase starts to react at later ages. This can be observed by following the main (non-overlapped) diffraction peak of C_2_S located at 12.4°. Finally, it must be highlighted that the main diffraction hump/peak of C–S–H gel, located at ~12°, is not present which indicates that the crystallinity of the gel (in these conditions) is very low. This peak is overlapped with the diffraction peak of calcite which allows one to rule out carbonation of the pastes and it is also an indirect evidence of the tightness of the system.

### 3.3. Quantitative Phase Analyses

Spinning of the capillary resulted in accurate powder diffraction intensities for the in situ recorded powder diffraction patterns. This statement is supported by the quality of the Rietveld fits, which make it possible to quantitatively determine the phase contents at the different hydration times; see below. Rietveld methodology, without the addition of an internal standard [24], does not allow determinations of the overall amount of amorphous and crystalline not-quantified (ACn) content to be made without assumptions. Here, it was decided not to add an internal standard because it was not known if (i) it could react with component(s) of the paste under high temperature and pressure, (ii) it could modify the kinetic of the reactions by providing additional surface for crystallization and precipitation (filler effect). Research is needed to establish whether an appropriate internal standard can be used under these (demanding) conditions. On the other and, external standard methodology is being widely used in reflection geometry. However, for capillary transmission geometry, the situation is more complex as the packing degree of the standard and the sample, within the capillaries, must be the same or known with precision. Furthermore, the time-evolution of the packing degree (under pressure and temperature) must be the same (or at least known). Again, research on to this subject is needed to determine whether external standard methodology can be used here.

RQPA have been carried out as described in the experimental section, i.e., following the same procedures developed for ex situ recorded data [24,25,26]. Figure 4 displays two Rietveld plots at selected hydration times, ~2 and ~11 h. The quality of the fits is very good as ensured by the flatness of the difference curves (blue lines at the bottom of each panel). Direct RQPA results are shown in Appendix A. In addition to the phase contents, that table also gathers the values for the Rietveld disagreement factor (R_WP_) [25] of all the fits. The values are low, indicating good agreement between the data and the model but R_WP_ values can also be low because the background values are relatively high. Therefore, Appendix A also reports the R_F_ values [25] for alite, which only depend upon the quality of the data and the appropriateness of the structural description. The low values of R_F_ also indicate the good quality of the in situ diffraction data.

As stated above, ACn contents cannot be determined from the acquired data without assumptions being made [27]. However, if one (or more) component phases do not react, they can be used as standard to determine the overall amount of ACn. Belite (C_2_S phase), which from Figure 1, Figure 2 does not seem to react up to 3 h and 15 min of hydration, was used as standard up to this hydration time. Hence, it is assumed here that the amount of belite do not change from t_0_ to 3 h and 15 min. Additionally, C_4_AF phase does not seem to react from 5 h and 20 min up to 14 h as their diffraction peaks do not decrease in intensity, see Figure 1 and Figure 3. Therefore, it is also assumed here that the content of C_4_AF do not change from 5 h and 20 min to the end of the experiment. A recent publication reported the low reactivity of this phase at high temperature and pressure [17]. Under these constraints, the ACn contents can be calculated; the full quantitative phase analyses are reported in Table 1 and displayed in Figure 5. For time = 0, ACn is just the added free water, as it is assumed here that the OWC does not contain any amorphous fraction. Furthermore, as this is a quantitative study, the remaining free water and the overall amount of amorphous solid content can be derived from ACn values after the chemically bounded water is estimated, see Table 1. The chemically bounded water was estimated from reactions (1)–(3) assuming a katoite stoichiometry of Ca_2.93_Al_1.97_(SiO_4_)_0.64_(OH)_9.44_.
(3)$$C_{2}S+4.2H\to 0.2\mathrm{CH}+{\mathrm{Ca}}_{1.8}{\mathrm{SH}}_{4}$$

Figure 5 and Table 1 show the quantitative evolution of the different component phases over time. Firstly, the evolution of the anhydrous phases will be discussed. Alite starts to react very quickly; at 9 min after reaching the selected pressure, a decrease in percentage is noticeable. After 7 h, its content has decreased by 90%, being below 4 wt%. Remarkably, belite starts to react after 5 h and the degree of reaction of this phase is ~42% after 14 h of hydration. To the best of our knowledge, the in situ reactivity of belite has not been reported in any OWC powder diffraction study, as their diffraction peaks are strongly overlapped with those of alite. As expected, the dissolution/reactivity of C_3_A is very fast and this phase is totally consumed in less than 1 h of hydration. Finally, the reactivity of C_4_AF phase is very peculiar. It reacts at early ages, see Table 1 and Figure 5, about 28% up to 3 h and then its reactivity is inhibited. The hydration of this phase under high temperature and pressure deserves more research.

The crystallization of portlandite starts ~30 min and its acceleration takes place between 1 and 4 h of hydration. At 14 h, the portlandite content was 12.9 wt%, see Table 1. The accuracy of these analyses can be estimated through a back of the envelope calculation following the chemical reactions described in the introduction. As the reactivity of alite and belite have been measured, the expected amount of the formed portlandite can be obtained. The resulting number is 16.0 wt% (15.6 wt% from C_3_S and 0.4 wt% from C_2_S). The theoretically expected portlandite content, 16 wt%, is larger than the measured value, 12.9 wt%, because portlandite is consumed by some of the crystalline products formed at high temperature and pressure and discussed below. Furthermore, the paste after this in situ study was extracted from the capillary, and a thermal analysis study was carried out to independently quantify the overall amount of portlandite. This study was carried out just after the synchrotron experiment to avoid/minimize carbonation. Figure 6 shows the thermal analysis of the resulting paste where the portlandite content was determined from the tangential method as recommended in the latest reference book [39]. The weight loss due to the water release from portlandite was 3.4 wt% which translates to an overall content of crystalline portlandite of 14.0 wt%. The relatively good agreement between the amount of portlandite quantified in the in situ experiment, 12.9 wt%, and the amount measured ex situ by thermal analysis, 14.0 wt%, which is also close to the expected value from alite and belite reactions, demonstrate the accuracy of these in situ analyses.

The alite reactivity determined in this study can be compared to those previously reported. Table 2 gathers our key results, which are compared to related studies, where the reactivities are extracted from the evolution of the area of single diffraction peaks. From the data reported in Table 2, it can be concluded that alite reactivity is faster in cements that in a pure alite sample, which is also the case at room temperature and atmospheric pressure. It can also be inferred that the temperature accelerates the hydration of alite more than pressure. Reports of 50% alite reactivity in 90–150 min are quite common which is the reason to use retards in field applications. Finally and very importantly, belite reactivity has not been mentioned in any work, even with belite contents being higher than 30 wt% for some cements.

A small amount of ettringite crystallizes at a very early age, and it decomposes after 26 min of hydration at the applied pressure and temperature. In addition, katoite starts to precipitate before 26 min of hydration and its concentration reaches its maximum, ~6.5 wt%, close to 3 h of hydration. Finally, the formation of the non-ambient hydrated phases such as Jaffeite and α-C_2_SH have also been quantified, see Table 1. The percentage of Jaffeite is very low, below 0.5 wt%, after 14 h of hydration. Conversely, the amount of crystalline α-C_2_SH is significant, close to 2 wt%. According to the literature [12], the formation of α-C_2_SH mainly occurs at the beginning of the cement hydration at high pressure and temperature conditions and then, this phase tends to convert into Jaffeite. These phases may form from the reaction of C–S–H gel and portlandite. These reactions could be further investigated by ex situ experiments; the synchrotron beamtime cannot be used to follow slow kinetics, as it would require too much time (in situ) experiments.

Table 1 and Figure 5 also report the ACn values. It is important to mention that the values obtained from this methodology encompass not only amorphous solid materials, but also the free water within the capillary and any crystalline phase not computed in the control file of the Rietveld fit. Figure 5 shows that ACn content increases with time due to the precipitation of large amounts of amorphous/nanocrystalline phases, mainly C–S–H gel. C–S–H gel can react at high pressure and temperature to yield crystalline phases, for instance Jaffeite, but this is not largely observed in the measured time window under the explored conditions.

## 4. Conclusions

The implementation of a new cell, specially designed to rotate under high pressure and temperature, currently up to 200 bars and 200 °C, made it possible to carry out in situ synchrotron powder diffraction studies of evolving powders. The quality of the powder diffraction data was sufficient to carry out Rietveld quantitative phase analyses in order to follow the reaction of each component in an OWC paste at 150 bars and 150 °C. Chiefly, shown the alite and belite reactivities were determined. For instance, the reaction degree of alite reaches 90% after 7 h and that of belite was 42% at 14 h. Furthermore, the measured crystalline portlandite content after 14 h, 12.9 wt%, was shown to be accurate, as it agrees fairly well with the amount expected from the reactivity of alite and belite and with the ex situ measurement by thermal analysis of the final product.

## Figures and Tables

**Figure 1 materials-12-01897-f001:**
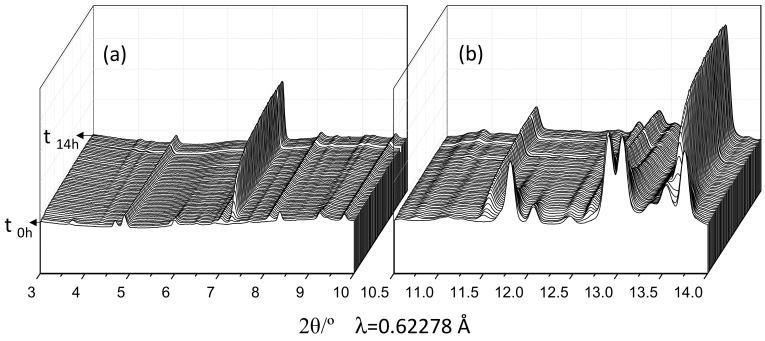
3D view of the SXRPD raw patterns for the OWC paste collected at 150 bars and 150 °C every 15 min of hydration up to 14 h. (**a**) Region from 3.0 to 10.0° (2θ) and (**b**) region from 10.5 to 14.0° (2θ).

**Figure 2 materials-12-01897-f002:**
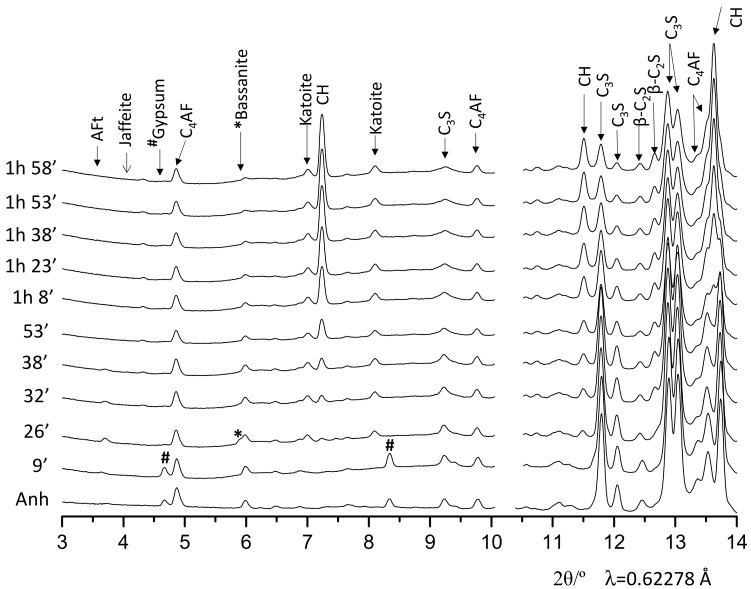
Selected range of SXRPD raw patterns for the OWC paste collected at 150 bars and 150 °C at early hydration ages (up to 2 h), with the main diffraction peaks due to a given phase labelled.

**Figure 3 materials-12-01897-f003:**
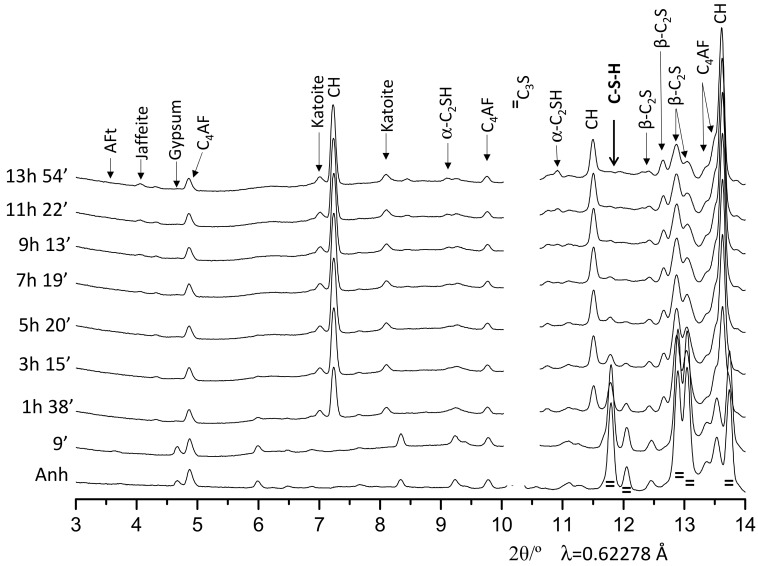
Selected range of SXRPD raw patterns for the OWC paste collected under 150 bars and 150 °C at late hydration ages (up to 14 h), with the main diffraction peaks due to a given phase labelled.

**Figure 4 materials-12-01897-f004:**
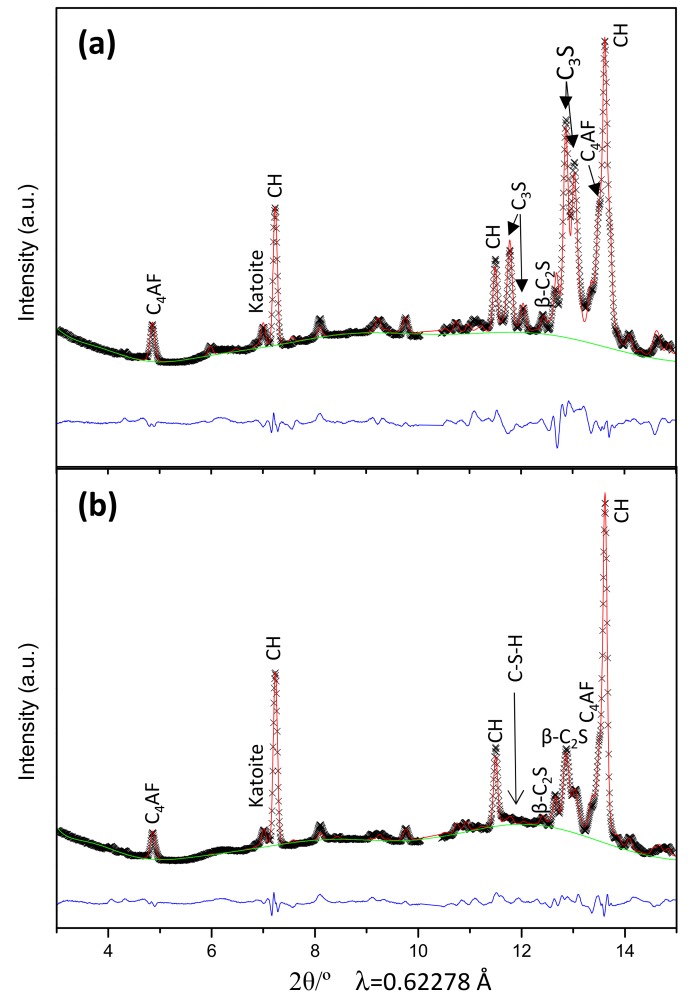
In situ Rietveld synchrotron powder X-ray diffraction plots for the oil well cement paste hydrating under 150 bars and 150 °C for (**a**) 1 h and 38 min, and (**b**) 11 h and 22 min.

**Figure 5 materials-12-01897-f005:**
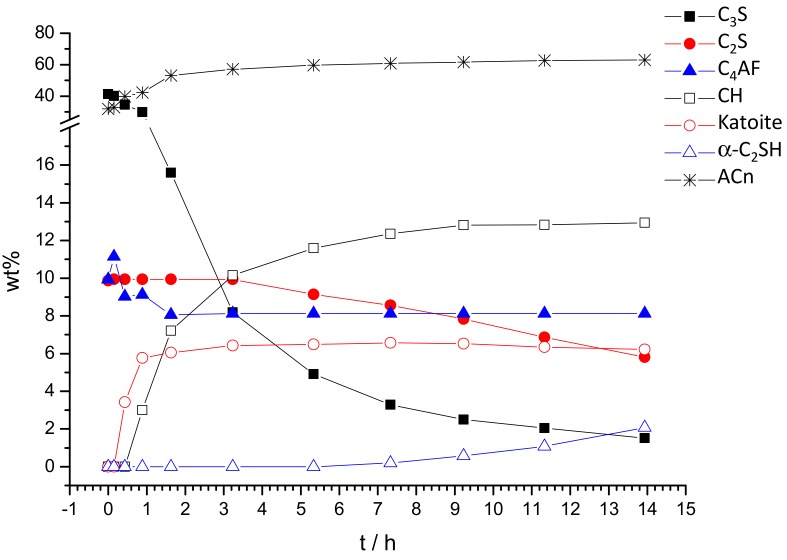
Re-normalized Rietveld quantitative phase analysis results for the OWC paste hydrated up to 14 h under 150 bars and 150 °C including the overall amorphous content.

**Figure 6 materials-12-01897-f006:**
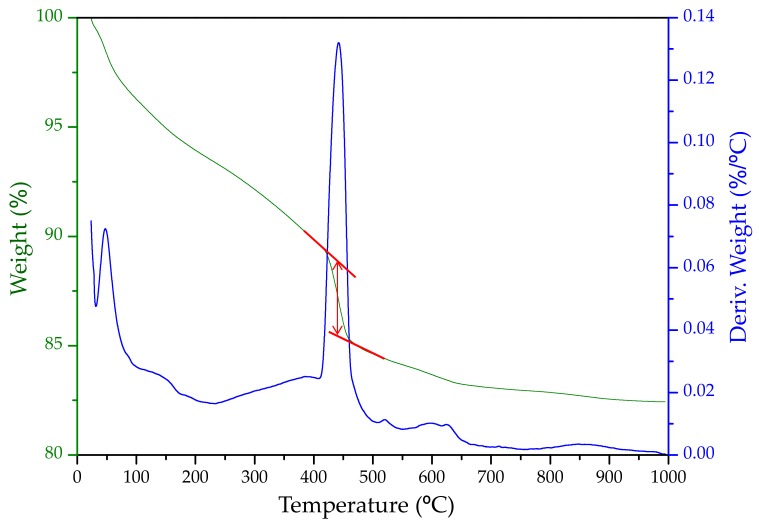
Ex situ thermal analysis data (weight loss—green curve, derivative of the weight loss—blue curve) for the final paste after the synchrotron experiment to quantify the overall crystalline portlandite content from the tangential method (showed in red, 3.4 wt%).

**Table 1 materials-12-01897-t001:** Re-normalized RQPA for the OWC paste hydrated under 150 bars and 150 °C. Amorphous content determination is also included, obtained by internal standard methodology, considering β-C_2_S as standard up to 3 h and 15 min and C_4_AF afterward. The numbers in bold highlight the assumption of no reaction of these phases in the reported time intervals.

Phase/wt%	t_0_	9′	26′	53′	1 h 38′	3 h 15′	5 h 20′	7 h 19′	9 h 13′	11 h 22′	13 h 54′
C_3_S [28]	41.6(1)	40.1(1)	34.6(1)	29.9(1)	15.6(2)	8.2(2)	4.9(3)	3.3(2)	2.5(3)	2.1(3)	1.5(3)
β-C_2_S [29]	9.9(2)	9.9(-)	9.9(-)	9.9(-)	9.9(-)	9.9(-)	9.1(3)	8.6(3)	7.8(3)	6.9(3)	5.8(3)
C_4_AF [30]	11.3(1)	11.1(1)	9.0(1)	9.1(1)	8.1(2)	8.1(-)	8.1(-)	8.1(-)	8.1(-)	8.1(-)	8.1(-)
o-C_3_A [31]	2.3(1)	2.2(1)	1.1(1)	0.0	0.0	0.0	0.0	0.0	0.0	0.0	0.0
C $\bar{S}$ H_2_ [32]	2.9(2)	3.1(1)	0.0	0.0	0.0	0.0	0.0	0.0	0.0	0.0	0.0
C $\bar{S}$ H_0.5_ [33]	0.0	0.0	1.1(1)	0.0	0.0	0.0	0.0	0.0	0.0	0.0	0.0
CH [34]	0.0	0.0	0.3(1)	3.0(1)	7.2(1)	10.1(1)	11.6(1)	12.4(1)	12.8(1)	12.8(1)	12.9(1)
AFt [35]	0.0	0.6(1)	0.6(1)	0.0	0.0	0.0	0.0	0.0	0.0	0.0	0.0
Jaffeite [36]	0.0	0.0	0.0	0.0	0.0	0.0	0.0	0.0	0.0	0.0	0.3(2)
Katoite [37]	0.0	0.0	3.5(1)	5.8(2)	6.0(2)	6.4(2)	6.5(2)	6.6(2)	6.5(2)	6.3(2)	6.2(2)
α-C_2_SH [38]	0.0	0.0	0.0	0.0	0.0	0.0	0.0	0.2(3)	0.6(3)	1.1(3)	2.1(3)
FW ^$^	32.0(-) ^#^	31.2	26.3	21.9	16.0	12.9	11.2	10.2	9.6	9.0	8.3
Amorph. ^#^		1.7	13.5	20.4	37.2	44.3	48.6	50.7	52.1	53.7	54.7

^$^ FW is estimated by subtracting the calculated chemically bounded water from the added water. ^#^ It is assumed that the OWC does not contains amorphous fraction and so the w/c mass ratio of 0.47 corresponds to 32 wt% of free water (47 g of water in 147 g of paste). The overall amorphous solid content is determined by subtracting the free water from the ACn values.

**Table 2 materials-12-01897-t002:** Summary of the hydration details including time for 50% alite reactivity [t_50%_(C_3_S)], time for 90% alite reactivity [t_50%_(C_3_S)] and time for 50% belite reactivity [t_50%_(C_2_S)]. n.r. stands for not reported.

Type of Sample	Blaine/m^2^/kg	w/c Mass Ratio	T/°C	P/Bars	t_50%_(C_3_S)/min	t_90%_(C_3_S)/h	t_50%_(C_2_S)/h	Remarks	Ref.
Class G-HSR	340	0.47	150	150	98	6.6	~15	-	This work
Pure-C_3_S	n.r.	0.44	160	400	n.a.	16	-	$	[11]
Class G	310	0.40	130	~5	160	~6	-	#	[40]
Class G	n.r.	0.412	80	400	150	>13	-	@	[6]
Class H	n.r.	0.38	57	“	“	-	-	&	[7]
Class A	327	n.r.	60	~1000	90	-	-	-	[10]

^$^ No kinetics data at early hydration ages are reported. ^#^ C_2_S or belite is not mentioned in the paper. ^@^ The cement contained 33 wt% of C_2_S but belite reactivity was not referred to. “ t_50%_ (C_3_S) was ~970, 600, 490, 380, 230, and 75 min for 330, 470, 600, 900, 1200 and 1930 bars; respectively. ^&^ The paste contained 0.1 wt% of maltodextrin retardant. The cement contained 25 wt% of C_2_S but belite reactivity was not mentioned.

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
