# Peer review of "Rietveld Quantitative Phase Analysis of Oil Well Cement: In Situ Hydration Study at 150 Bars and 150 °C"

_materials, 2019, doi:10.3390/ma12121897_

Round 1
Reviewer 1 Report
The manuscript titled in “Rietveld quantitative phase analysis of Oil Well Cement: in situ hydration study at 150 bars and 150 degC” addressed the hydration behavior of oil well cement at high temperature and pressure based on the results of their phase assemblage calculated from Rietveld refinements. The refinements were moderately-well conducted and the topic was also interesting. There was a few issues for potential publication of this manuscript in Materials journal.
1. During cement hydration, liquid H2O transfers to solid-state H2O as chemically bound water in hydration products, implying that the total solid amounts of XRD specimens increases with time. So, the authors have to normalize their Rietveld quantification results with chemically bound water content to compare the samples measured at different ages.
2. The authors calculated ACns with relatively less soluble phase’s amount. Although they explained why internal standard method was not adopted in this study, I guess, the authors could use external standard method for ACn calculation because synchrotron-based XRD would be well calibrated and its intensity could be enough-well adjusted. The authors have to explain why they did not use the external standard method for this study.
3. The results were moderately-well described, but the discussions of the results were relatively feeble. It is recommended that the authors compare their results with relevant references of OPC and OWC hydration study at different curing condition in order to emphasize the novelty of this study to readers.
Author Response
The response to the reviewer comments are described in the attached letter.

Reviewer 2 Report
The paper "Rietveld quantitative phase analysis of Oil Well Cement: in situ hydration study at 150 bars and 150ºC" reports an original and well conducted research on phase reactivities ans quantitative phase analysis of oil and gas well cements.
The paper is well written in all its parts, the experimental section is complete and clear, as all the technique are widely described, and the results are clearly displayed and reported.
I would recommend to accepr this paper after some minor revisions that are listed below.
abstract:
-line 12: change "Its overall reactivity" with "Their overall reactivity"
Table 1: please add the sigmas of the weight fractions.
Table S2: please add the sigmas of the weight fractions.
Figure S3: what are the spots in all the three pictures? Are the sapphire diffraction spots? Should be indicated...
Author Response

(The authors gave the same response as above.)
